# Teclistamab Dosing Strategies in Relapsed/Refractory Myeloma: A Real-World Comparison of Weekly and Biweekly Versus Fixed Intervals

**DOI:** 10.3390/cancers17213569

**Published:** 2025-11-05

**Authors:** Jordan Snyder, Shebli Atrash, Barry Paul, Abdullah Mohammad Khan, Alma Habib, Hira Shaikh, Christopher Strouse, Omar Alkharabsheh, Anita Mazloom, Nausheen Ahmed, Zahra Mahmoudjafari, Muhammad Umair Mushtaq, Anas Zayad, Joseph McGuirk, Yun Kyoung Tiger, Mansi R. Shah, Al-Ola Abdallah

**Affiliations:** 1Division of Hematologic Malignancies & Cellular Therapeutics, University of Kansas Medical Center, Westwood, KS 66205, USA; 2Levine Cancer Center, Atrium Health, Wake Forest University School of Medicine, Charlotte, NC 27103, USA; 3Division of Hematology, The Ohio State University Comprehensive Cancer Center, Columbus, OH 43210, USA; 4Division of Hematology, Oncology, and Blood & Marrow Transplantation, University of Iowa, Iowa City, IA 52242, USA; 5Division of Hematology and Oncology, Mitchell Cancer Institute, University of South Alabama, Mobile, AL 36604, USA; 6Department of Internal of Medicine, Hamad Medical Corporation, Doha P.O. Box 3050, Qatar; 7Division of Blood Disorders, Rutgers Cancer Institute, New Brunswick, NJ 08903, USA

**Keywords:** teclistamab, relapsed/refractory multiple myeloma, dosing strategies

## Abstract

**Simple Summary:**

Patients with multiple myeloma who relapse after several treatments often receive a new immune-based therapy called teclistamab. This drug is usually given continuously until the disease worsens or side effects become intolerable. However, long-term treatment can increase infection risk, lower blood counts, and reduce quality of life. In real-world practice, doctors sometimes stop treatment after patients achieve a deep and stable response, but this approach has not been formally studied. In this study, we compared patients who continued treatment with teclistamab to those who stopped after responding well. We found that stopping treatment in carefully selected patients did not shorten survival or increase the risk of relapse in the first year. However, infections remained common, even after stopping treatment. These results suggest that fixed-duration therapy may be safe for some patients and should be evaluated further in clinical trials.

**Abstract:**

**Background/Objectives:** Teclistamab is a bispecific antibody targeting BCMA and CD3, approved for relapsed/refractory multiple myeloma. It is administered continuously until progression or intolerance; however, prolonged use may increase infections and treatment burden. This study compares continuous versus fixed-duration teclistamab to determine whether treatment discontinuation after response is feasible without compromising outcomes. **Methods:** A multicenter retrospective study was conducted on adults with relapsed/refractory multiple myeloma treated with teclistamab between August 2022 and May 2024. Patients received step-up dosing followed by weekly administration. Those who achieved ≥VGPR and discontinued therapy due to deep response, toxicity, or preference were assigned to the fixed-duration group. Outcomes included response rates, progression-free survival (PFS), overall survival (OS), and adverse events. **Results:** Eighty-eight patients were included (continuous: *n* = 72; fixed: *n* = 16). The fixed group had higher complete response rates (69% vs. 44%) and shorter median time to best response (1 vs. 2 months). Median PFS was 16 months for continuous dosing versus 13 months for fixed-duration. Twelve-month PFS was similar (65% vs. 66%). Twelve-month OS was 83% vs. 81% in the continuous and fixed groups, respectively. Cytokine release syndrome and neurotoxicity rates were similar. Infections were more frequent and severe in the fixed cohort (75% any grade; 69% grade ≥ 3). **Conclusions:** Fixed-duration teclistamab after deep response appears feasible in appropriately selected patients, with comparable early survival outcomes to continuous treatment. Prospective studies are needed to define selection criteria, immune recovery markers, and optimal discontinuation timing.

## 1. Communication

Multiple myeloma (MM) is a hematological cancer characterized by proliferation of malignant plasma cells in the bone marrow [1,2]. Relapsed/refractory multiple myeloma (RRMM) poses significant treatment challenges due to drug resistance and disease variability [3]. Teclistamab, a BCMA-targeting bispecific antibody, has shown promise in improving RRMM outcomes. In the MajesTEC-1 trial with 165 patients, there was a 63% overall response rate (ORR) and a 39.4% complete response (CR), with a median progression-free survival (PFS) of 11.3 months and overall survival (OS) of 18.3 months. Common adverse events included neutropenia (70.9%), anemia (52.1%), and cytokine release syndrome (72.1%). Teclistamab is initially dosed at 1.5 mg/kg weekly, with possible biweekly adjustments after six months of CR [4,5].

In the MajesTEC-1 trial, patients switching from weekly to biweekly teclistamab dosing had a median response duration of 20.5 months. Flexible dosing based on patient preferences can improve adherence, satisfaction, and clinical outcomes [6,7]. We investigate individualized dosing strategies, focusing on fixed dosing for responders in RRMM patients. However, while recent real-world and pharmacokinetic data indicate that de-escalation to biweekly dosing maintains disease control and reduces infection burden and treatment fatigue, no randomized or head-to-head studies have yet compared weekly, biweekly, and fixed-duration regimens [8,9]. Our study assesses the efficacy of fixed-dose teclistamab compared to continuous weekly or biweekly dosing. Patients who achieve at least a partial response and experience significant adverse events will stop treatment after a set period and will be monitored without ongoing therapy. This approach reflects real-world clinical decision-making, aiming to balance efficacy and quality of life and minimize treatment-related burdens [9,10]. Long-term continuous therapy is associated with treatment fatigue, frequent hospital visits, and impaired quality of life in RRMM patients, which has led to increasing interest in response-adapted or time-limited treatment strategies [11].

We conducted a retrospective chart review of RRMM patients treated with teclistamab between August 2022 and May 2024.

Patient selection followed criteria adapted from the pivotal MajesTEC-1 and MajesTEC-2 trials [4,12]. Eligible patients were adults (≥18 years) with confirmed multiple myeloma based on IMWG diagnostic criteria and measurable disease at screening (serum M-protein ≥ 1 g/dL, or urine M-protein ≥ 200 mg/24 h, or abnormal free-light-chain ratio ≥ 10 mg/dL). All had relapsed or refractory multiple myeloma (RRMM) after prior exposure to at least one proteasome inhibitor (PI) and one immunomodulatory drug (IMiD) Participants were required to have an adequate hematologic, renal, and hepatic function. Major exclusion criteria included prior exposure to BCMA-targeted therapy, recent live vaccination, receipt of high-dose corticosteroids within 14 days, active CNS involvement, or HIV seropositivity.

For clarity, RRMM was defined per IMWG as disease that progressed during or within 60 days after the last therapy or failed to achieve a minimal response [13]. High-risk cytogenetics were defined by the presence of del(17p), t(4;14), t(14;16), t(14;20), or 1q gain/amplification, following IMWG/EMN standards [14].

Teclistamab was administered according to institutional practice consistent with the FDA-approved regimen and MajesTEC-1 protocol [4]. Treatment began with subcutaneous step-up doses of 0.06 mg/kg, 0.3 mg/kg, and 1.5 mg/kg on days 1, 4, and 7, respectively, followed by a maintenance dose of 1.5 mg/kg once weekly (QW) until disease progression or unacceptable toxicity. In selected patients who achieved and maintained a confirmed complete response (CR) for ≥6 months, the dosing interval extended to every 2 weeks (Q2W).

The fixed-duration approach, not formally evaluated in the pivotal trials, was applied in real-world practice when treatment was discontinued after a sustained deep response (≥VGPR) or in the setting of clinically significant toxicity. After discontinuation, patients underwent clinical and laboratory monitoring every 4 weeks until progression.

We summarized the demographic data and prognostic factors using chi-squared or Fisher tests for categorical variables and the Wilcoxon Mann–Whitney test for quantitative variables. PFS and OS were calculated from the start of treatment. Of the 181 identified patients (165 continuous dosing and 16 fixed dosing), we excluded 91 with less than 3 months of response or follow-up, resulting in an analysis of 88 patients (72 continuous dosing and 16 fixed dosing).

The median age of participants was 69 years (range: 64–73) for the continuous cohort and 73 years (range: 70–84) for the fixed cohort, with 56% and 50% male, respectively. Both groups had a median of 5 prior therapy lines (range: 4–7). In total, 81% were triple-class refractory, with 51% (continuous) and 56% (fixed) being penta-class refractory. BCMA-refractoriness was observed in 29% of patients in the continuous group and 56% of patients in the fixed group. High-risk cytogenetics were present in 43% (continuous) and 44% (fixed) of patients, while extramedullary disease occurred in 21% and 19% of patients, respectively. ECOG performance status of 2 or higher was noted in 18% (continuous) and 31% (fixed). Additionally, 36% of the continuous cohort and 56% of the fixed cohort would have been ineligible for the MajesTEC-1 trial (Table 1).

By design, all patients achieved at least a partial response. In the continuous treatment cohort, 44% achieved a CR or better, whereas this rate was 69% in the fixed cohort. The median time to first response was 29 days (range, 5–270 days) in the continuous cohort and 15 days (range: 7–30) in the fixed cohort. The median time to the best response was 2 months (range: 0.5–11) for continuous and 1 month (range: 0.5–3) for fixed. After a median follow-up of 12 months (IQR 6.0–15.0), the median progression-free survival (PFS) was 16 months in the continuous cohort and 13 months in the fixed cohort (hazard ratio [HR] 1.7; 95% CI 0.77–3.75; *p* = 0.20). The corresponding 12-month PFS estimates were 65% (95% CI 54–77%) and 66% (95% CI 45–96%) for the continuous and fixed groups, respectively. The median overall survival (OS) was not reached in the continuous cohort compared with 14 months in the fixed cohort (HR 2.79; 95% CI 0.96–8.07; *p* = 0.048). Although Kaplan–Meier curves visually appeared to favor the fixed-duration cohort for early disease control, these differences did not reach statistical significance. As in Figure 1 and Figure 2.

In the fixed cohort of 16 patients, all patients received at least a step-up dose during hospitalization, with no ICU admissions. The reasons for discontinuation included severe infection (56%), delayed neurotoxicity (25%), patient preference (12%), and a new diagnosis of gastric cancer (6%). In this cohort, five patients (31%) died of progressive disease (3 patients [19%]), COVID-19 (1 patient [6%]), and Clostridium difficile infection (1 patient [6%]) unrelated to teclistamab treatment. Currently, seven patients (44%) are in remission and off treatment for an average of 12.6 months (range: 2–20.6). Overall, 38% of the patients off treatment maintained a treatment-free status for over 12 months.

The median duration of treatment for all patients in the fixed cohort was 2.7 months (range: 0.5–8.5). In contrast, the median duration of treatment cessation until progression was 8.2 months (range: 2–20.6), as illustrated in Figure 3. At data cutoff, seven patients (44%) experienced disease progression while off treatment, with an average time to treatment progression of 7.6 months (range: 3.1–14.8), while seven (44%) remained in remission without therapy. Among the patients who progressed, four were re-challenged with teclistamab, two began treatment with elranatamab, and one patient was treated with pomalidomide and dexamethasone.

In comparing adverse events (AEs) between the continuous and fixed dosing cohorts of teclistamab, both showed similar toxicity profiles, but notable differences emerged. Cytokine release syndrome (CRS) occurred at comparable rates (54% in continuous vs. 57% in fixed), with fixed dosing reporting no grade 3/4 CRS cases. Immune effector cell-associated neurotoxicity syndrome (ICANS) rates were also similar (5.6% continuous vs. 6.3% fixed), with no grade 3/4 cases in the fixed group.

However, the fixed cohort had a higher frequency of infections (75% vs. 59%) and grade 3/4 infections (69% vs. 22%, *p* < 0.001). Elevated liver enzyme levels were also more common in the fixed group (25% vs. 15%), although none resulted in grade 3/4 events. Hematological toxicities assessed on day 30, such as leukopenia and neutropenia, were slightly less common in the fixed group (Table 2).

Overall, supportive care interventions were similar, with granulocyte colony-stimulating factor administered to 22% of the continuous cohort and 19% of the fixed cohort (*p* > 0.9). Intravenous immunoglobulin (IVIG) was more common in the continuous cohort (75% vs. 56%, *p* = 0.2), and packed red blood cell transfusions were required for 22% of the continuous group and 13% of the fixed group (*p* = 0.5). Dose delays were more frequent in the fixed cohort (75%) than in the continuous cohort (53%), although the difference was not statistically significant (*p* = 0.1). These findings suggest that supportive care needs are similar across dosing strategies, with a possible trend toward more treatment interruptions in the fixed-dose group.

Although CRS and ICANS rates were comparable between cohorts, the higher infection incidence in the fixed-duration group may reflect long-term immune suppression rather than acute cytokine toxicity [15]. Teclistamab and other BCMA-directed bispecific antibodies cause prolonged hypogammaglobulinemia and B-cell depletion, leading to impaired humoral immunity and a sustained risk of bacterial and viral infections [16]. Independent real-world cohorts have confirmed that teclistamab-treated patients remain vulnerable to severe and late-onset infections due to delayed humoral and cellular immune reconstitution, even after therapy discontinuation [17]. In addition, prolonged cytopenias, particularly neutropenia and lymphopenia, have been reported with teclistamab and may persist beyond the treatment window, contributing to infection susceptibility [15,18]. Furthermore, continuous T-cell activation may induce functional exhaustion, impairing immune surveillance even after treatment discontinuation [19]. The fixed-duration cohort also included patients with more prior lines of therapy and longer observation periods, increasing cumulative infection exposure [20]. Differences in prophylactic practices, such as intravenous immunoglobulin replacement or antimicrobial prophylaxis, may have further influenced infection reporting [21]. These mechanisms together may explain the imbalance in infection rates despite similar CRS and ICANS frequencies.

In this multicenter retrospective analysis of patients with RRMM, we assessed the outcomes and safety of fixed versus continuous teclistamab dosing. Our findings suggest that fixed-duration treatment is feasible and potentially effective in a selected patient population while highlighting important safety trade-offs. In clinical practice, the decision between continuous, biweekly, and fixed-duration teclistamab administration is individualized according to disease response, treatment tolerance, and patient preference. Continuous weekly dosing remains the standard of care and is preferred for patients with ongoing disease activity or partial response [4]. Transition to a biweekly schedule is generally reserved for patients achieving a confirmed complete response sustained for at least six months, in line with FDA labeling [22].

Selection of patients for a fixed-duration teclistamab strategy must be cautious and individualized. The most suitable candidates may be those who achieve sustained deep responses (≥VGPR), without biochemical relapse, and who demonstrate or are expected to demonstrate recovery of immune parameters, such as stable or increasing immunoglobulin levels or lymphocyte counts. This approach may also be explored in patients who experience persistent cytopenias, recurrent infections, or declining treatment tolerance. Recent studies suggest that in patients achieving sustained deep responses, treatment discontinuation may be feasible without compromising disease control, supporting ongoing evaluation of fixed-duration teclistamab strategies [23]. However, patients with aggressive disease biology or high-risk cytogenetics should generally continue continuous dosing until progression. These considerations are informed by emerging real-world experience with teclistamab, which underscores the complexity of immune recovery and infection risk after prolonged therapy [24].

The fixed cohort had a higher complete response rate of 69% compared to 44% in the continuous dosing cohort, with a median time to best response of 1 month vs. 2 months, largely due to patient selection aimed at achieving deeper responses and other treatment discontinuation factors. Our observations of the duration of response in the fixed dosing cohort align with previous data suggesting that the early activity of teclistamab can be both profound and durable. Nearly half of patients in the fixed dosing cohort remained in remission off treatment, with 38% maintaining a treatment-free interval over 12 months, suggesting a potential benefit in therapy burden and quality of life [25]. In terms of safety, both cohorts had similar CRS rates, with no grade 3/4 CRS in the fixed group. However, infections were more frequent and severe in the fixed cohort (75% overall, 69% grade ≥ 3) than in the continuous cohort (59% overall, 22% grade ≥ 3) (*p* < 0.001), possibly because of selection bias. Nevertheless, the observed infection burden emphasizes the importance of close monitoring and individualized treatment adjustments.

Hematologic toxicities, including neutropenia, anemia, and thrombocytopenia, were common but generally comparable between cohorts, with slightly fewer grade 3/4 events in the fixed group. These findings are consistent with the known safety profile of teclistamab observed in the MajesTEC-1 trial, suggesting that fixed dosing does not substantially alter hematologic toxicity risk when applied selectively [4,26].

The efficacy and safety of teclistamab are being evaluated with other myeloma-directed therapies. The MajesTEC-2 trial reported an 88.5% ORR in RRMM patients, but 92.6% experienced infections, commonly upper respiratory infections (44%) and pneumonia (29.6%) [27,28]. The RedirecTT-1 trial showed similar high efficacy with teclistamab and talquetamab but had a 15% mortality rate from treatment-related infections [29].

This study provides important evidence regarding the dosing flexibility of teclistamab. As BCMA-targeted therapies become more common, optimizing treatment schedules is crucial for better efficacy and tolerability. Durable responses to therapy could significantly impact RRMM management, particularly for patients facing adverse effects or treatment fatigue. Further trials on fixed or response-adapted dosing strategies are necessary to validate these findings and develop evidence-based guidelines.

Safety concerns continue to hinder long-term use. This study has several limitations, including its retrospective and non-randomized design, the small number of patients in the fixed-duration cohort, variability in follow-up duration, reliance on chart-based documentation for adverse event reporting, and the influence of physician judgment on treatment discontinuation decisions. Collectively, these factors could introduce potential bias, such as selection bias or an overestimation of efficacy in the fixed-duration group. Future prospective, randomized studies are needed to validate these findings and to better define the optimal duration of teclistamab therapy in multiple myeloma.

## 2. Conclusions

In this real-world analysis, both continuous and fixed-duration teclistamab strategies demonstrated meaningful clinical activity in patients with relapsed or refractory multiple myeloma. Continuous weekly or biweekly dosing remains the current standard, whereas fixed-duration discontinuation, when applied to patients with deep and durable responses, appears feasible and may lessen cumulative toxicity and infection burden in selected patients.

## Figures and Tables

**Figure 1 cancers-17-03569-f001:**
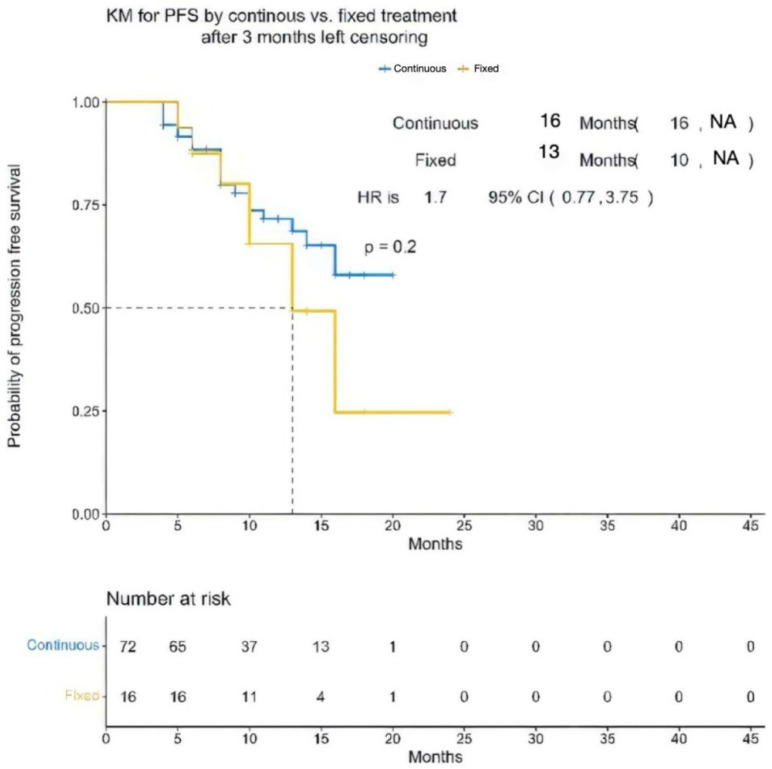
Kaplan–Meier progression-free survival (PFS) between patients receiving continuous dosing (n = 72) and fixed-interval dosing (n = 16). Median PFS was 16 months in the continuous cohort and 13 months in the fixed-duration cohort. The hazard ratio (HR) for progression or death was 1.7 (95% CI: 0.77–3.75; *p* = 0.20), indicating no statistically significant difference. NA; indicates not reached.

**Figure 2 cancers-17-03569-f002:**
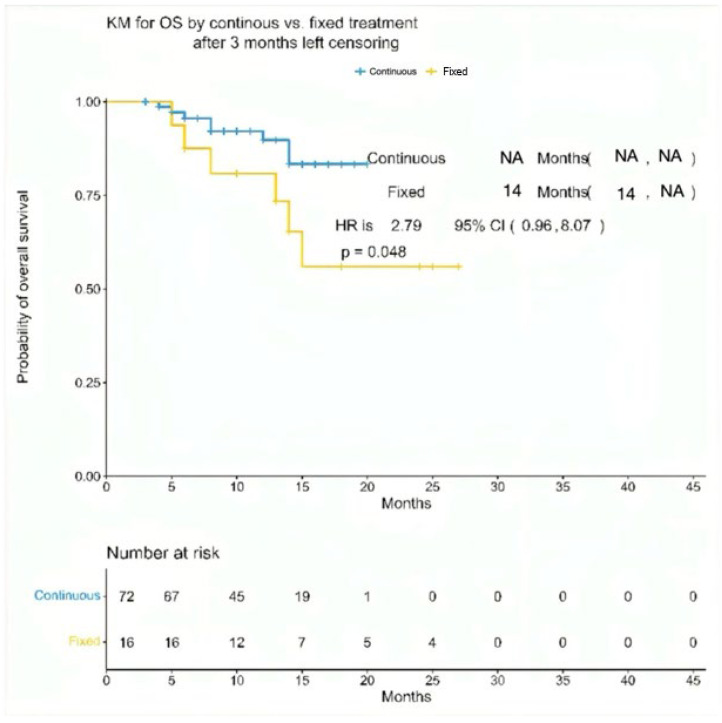
Kaplan–Meier overall survival (OS) between the continuous dosing cohort (n = 72) and the fixed-interval cohort (n = 16). Median OS was not reached (NA) in the continuous group and 14 months in the fixed-duration group. The hazard ratio (HR) for death was 2.79 (95% CI: 0.96–8.07; *p* = 0.048).

**Figure 3 cancers-17-03569-f003:**
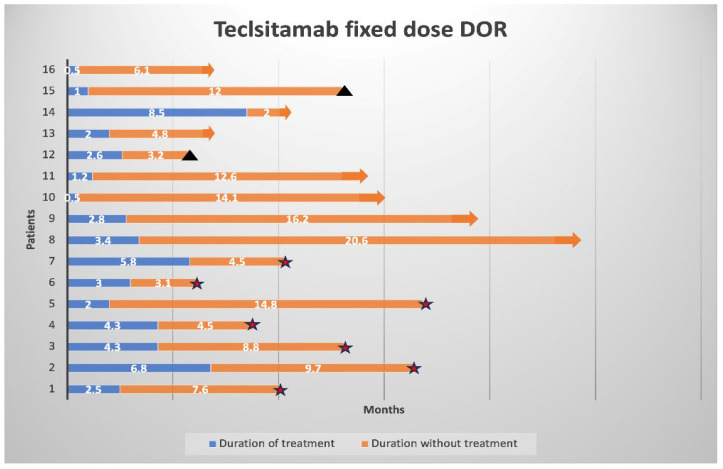
Duration of response (DoR) in patients receiving fixed-duration teclistamab. Each bar represents an individual patient (n = 16). Blue segments indicate the duration of teclistamab therapy in months, and orange segments indicate the period of remission without treatment in months. Black triangles mark death, and stars denote progression. Arrows indicate patients who remained in remission at the time of data cutoff.

**Table 1 cancers-17-03569-t001:** Patient characteristics in those who received teclistamab fixed vs. continuous:.

Characteristic	Overall (N = 88)	Teclistamab (Continuous Cohort) (N = 72)	Teclistamab (Fixed Cohort) (N = 16)	*p*-Value
Age, Median (Range)	71 (64–84)	69 (64–73)	73 (70–84)	0.018
Gender, n (%)				0.7
Female	40 (45)	32 (44)	8 (50)	
Male	48 (55)	40 (56)	8 (50)	
Race, n (%)				0.7
Caucasian	73 (83)	60 (83)	13 (81)	
African American	11 (13)	9 (13)	2 (13)	
Asian	2 (2)	2 (3)	0	
Hispanic	2 (2)	1 (1)	1 (6)	
Isotype, n (%)				0.3
IgG	49 (56)	39 (54)	10 (63)	
Non-IgG	18 (20)	16 (22)	2 (13)	
Light chain only	21 (24)	17 (243)	4 (25)	
R-ISS stage, n (%)				0.6
I	18 (20)	12 (17)	5 (31)	
II	41 (47)	34 (47)	7 (44)	
III	18 (20)	15 (21)	3 (19)	
Unknown	12 (14)	11 (15)	1 (6)	
* High-risk cytogenetics, n (%)	38 (43)	31(43)	7 (44)	>0.9
Extramedullary disease, n (%)	18 (20)	15 (21)	3 (19)	>0.9
Prior lines of therapy Median (range)		5 (4–7)	5.5 (4.5–7.5)	>0.9
ECOG PS, n (%)				0.14
0–1	70 (80)	59 (82)	11 (69)	
≥2	18 (20)	13 (18)	5 (31)	
Prior therapies exposed, n (%)				
Bortezomib	86 (98)	71 (99)	15 (94)	0.3
Carfilzomib	79 (90)	64 (89)	15 (94)	>0.9
PI exposure	88 (100)	72 (100)	16 (100)	>0.9
Lenalidomide	86 (98)	70 (97)	16 (100)	>0.9
Pomalidomide	82 (93)	67 (93)	15 (94)	>0.9
IMiD exposure	88 (100)	72 (100)	16 (100)	>0.9
Anti-CD38 MoAb	88 (100)	72 (100)	16 (100)	>0.9
ASCT	55 (63)	46 (64)	9 (56)	0.6
Refractory status, n (%)				
Bortezomib	62 (70)	50 (69)	12 (75)	0.8
Carfilzomib	71 (80)	58 (81)	13 (81)	>0.9
PI refractory	75 (85)	61 (85)	14 (88)	>0.9
Lenalidomide	68 (77)	55 (76)	13 (81)	>0.9
Pomalidomide	76 (86)	61 (85)	15 (94)	0.7
IMiD refractory	78 (89)	63 (88)	15 (94)	0.7
Anti-CD38 MoAb	84 (95)	68 (94%	16 (100)	>0.9
Double refractory	75 (85)	62 (86)	13 (81)	0.7
Triple Exposed	87 (99)	71 (99)	16 (100)	>0.9
Triple Refractory	71 (81)	58 (81)	13 (81)	>0.9
Penta Exposed	71 (81)	57 (79)	14 (88)	0.7
Penta Refractory	46 (52)	37 (51)	9 (56)	0.7
BCMA ADC (Belantamab)	15 (17)	11 (15)	4 (25)	0.5
* BCMA CAR-T cell	24 (27)	17 (26)	7 (43)	0.2
BDT exposed	35 (40)	26 (36)	9 (56)	0.14
BDT refractory	30 (34)	21 (29)	9 (56)	0.039

High-risk cytogenetics: TP53 mutation/17P del, 1q abnormality, t(4;14), t(14;16); ASCT: autologous stem cell transplantation; IMiD: immunomodulatory agent; PI: proteasome inhibitor; Anti-CD38 MoAb: Daratumumab or isatuximab; double refractory: refractory to both IMiD and PI; triple exposed/Refractory: exposed/refractory to at least one agent of each of the following class: IMiD, PI, Anti- CD38-MoAB; Penta exposed/refractory: exposed/refractory to 2 IMiDs, 2 PIs and one Anti- CD38-MoAB; ECOG PS, Eastern Cooperative Oncology Group performance status; R-ISS, Revised International Staging System. * BCMA CAR-Ide-cel and Cilta-cel were included, along with others from BCMA CAR-T clinical trials.

**Table 2 cancers-17-03569-t002:** Toxicity of teclistamab: continuous versus fixed.

Adverse Events (AEs) n (%)	Teclistamab Continuous Cohort (N = 72)		Teclistamab Fixed Cohort (N = 16)		*p*-Value
	Any Grades	Grade 3/4	Any Grades	Grade 3/4	
CRS	39 (54)	2 (3)	9 (57)	0	0.8
ICANS					
Elevated Liver function enzymes	11 (15)	1 (1)	4 (25)	0	0.4
Neutropenic fever	4 (6)	4 (6)	1 (6)	1 (6)	>0.9
Infection	42 (59)	16 (22)	12 (75)	11 (69)	<0.001
* Hematological AE					
Leukopenia	26 (36)	4 (6)	5 (32)	3 (19)	0.14
Neutropenia	18 (25)	8 (11)	5 (32)	4 (26)	0.2
Anemia	36 (49)	3 (4)	9 (56)	0	0.5
Thrombocytopenia	29 (40)	11 (15)	6 (38)	2 (13)	0.5

* All hematological toxicities were addressed on day 30 for comparison. Abbreviations: CRS—Cytokine Release Syndrome; ICANS—Immune Effector Cell-Associated Neurotoxicity Syndrome; AE—Adverse Event; L.

## Data Availability

The data generated and analyzed in this study are not publicly available due to privacy and ethical restrictions. De-identified data may be obtained from the corresponding author upon reasonable request and approval by the institutional ethics committee.

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
