# Peer review of "Teclistamab Dosing Strategies in Relapsed/Refractory Myeloma: A Real-World Comparison of Weekly and Biweekly Versus Fixed Intervals"

_cancers, 2025, doi:10.3390/cancers17213569_

Round 1
Reviewer 1 Report
Comments and Suggestions for Authors
This retrospective study aimed to compare the efficacy and safety of continuous teclistamab administration versus a fixed-duration approach in selected patients who achieved at least a partial response. The fixed-duration strategy, which involves treatment discontinuation in the setting of a meaningful response and/or significant adverse events, reflects a real-world, patient-centered approach intended to balance clinical efficacy with quality of life. Despite limitations related to the non-randomized design and the small number of patients in the fixed-dose group (n=16), the findings suggest that this approach may be feasible in a well-selected subset of patients. Notably, the fixed-dose group showed a higher complete response (CR) rate compared to the continuous group (69% vs. 44%), and a shorter time to response onset. However, these results must be interpreted cautiously, as the study design inherently selected patients with documented responses, and the decision to discontinue treatment was based on clinical judgment, introducing potential selection bias. Key limitations of this study include its retrospective design, small sample size (especially in the fixed-dose group), and the influence of clinical judgment on dosing decisions. Variability in follow-up durations and reliance on medical chart documentation for adverse event reporting further limit the interpretability of the findings. Additionally, treatment discontinuation due to toxicity in the fixed-dose cohort may have artificially prolonged the treatment-free interval in some cases, potentially overestimating the strategy’s efficacy. Despite these limitations, the findings suggest that fixed-duration teclistamab therapy—when applied selectively—may offer a valid option for certain patients by better balancing efficacy, tolerability, and quality of life.
No further comments.
Author Response
Reviewer 1
Comment:
This retrospective study aimed to compare the efficacy and safety of continuous teclistamab administration versus a fixed-duration approach in selected patients who achieved at least a partial response. The fixed-duration strategy, which involves treatment discontinuation in the setting of a meaningful response and/or significant adverse events, reflects a real-world, patient-centered approach intended to balance clinical efficacy with quality of life. Despite limitations related to the non-randomized design and the small number of patients in the fixed-dose group (n=16), the findings suggest that this approach may be feasible in a well-selected subset of patients. Notably, the fixed-dose group showed a higher complete response (CR) rate compared to the continuous group (69% vs. 44%), and a shorter time to response onset. However, these results must be interpreted cautiously, as the study design inherently selected patients with documented responses, and the decision to discontinue treatment was based on clinical judgment, introducing potential selection bias. Key limitations of this study include its retrospective design, small sample size (especially in the fixed-dose group), and the influence of clinical judgment on dosing decisions. Variability in follow-up durations and reliance on medical chart documentation for adverse event reporting further limit the interpretability of the findings. Additionally, treatment discontinuation due to toxicity in the fixed-dose cohort may have artificially prolonged the treatment-free interval in some cases, potentially overestimating the strategy’s efficacy. Despite these limitations, the findings suggest that fixed-duration teclistamab therapy—when applied selectively—may offer a valid option for certain patients by better balancing efficacy, tolerability, and quality of life.
Response 1:
Thank you for this observation. We have added a new paragraph at the end of the Discussion (page 11, line 5) to clearly acknowledge these limitations.
“This study has several limitations, including its retrospective and non-randomized design, the small number of patients in the fixed-duration cohort, variability in follow-up duration, reliance on chart-based documentation for adverse event reporting, and the influence of physician judgment on treatment discontinuation decisions. Collectively, these factors could introduce potential bias, such as selection bias or an overestimation of efficacy in the fixed-duration group. Future prospective, randomized studies are needed to validate these findings and to better define the optimal duration of teclistamab therapy in multiple myeloma.”

Reviewer 2 Report
Comments and Suggestions for Authors
The outlook of patients with relapsed and refractory myeloma is poor. Bispecific antibodies like Teclistamab may break the refractoriness of myeloma by combining antibody directing T cells to the tumor cells. Activation of T cells leads to the release of cytokines and ICANs. The reason why a continuous schedule is compared to a fixed schedule is not known. Maybe the tolerability is the reason for varying the schedule. However that seems to lead to the exclusion of large proportion of the continuous group. This selects the fittest patients and causes a misbalance. The figures are poorly explained: free from progression and overall survival appears better for the "fixed schedule group" although this is not written in the text. The flow diagram of duration of response seems only for the 16 patients of the fixed group, the triangles and the stars are not explained. Finally the higher incidence of infections in the fixed group is not explained, since the CRS and ICANS were not different. A valid study should include the given dose and schedule and the reasons for drop outs.
Author Response
Reviewer 2
Comment:
The outlook of patients with relapsed and refractory myeloma is poor. Bispecific antibodies like Teclistamab may break the refractoriness of myeloma by combining antibody directing T cells to the tumor cells. Activation of T cells leads to the release of cytokines and ICANs. The reason why a continuous schedule is compared to a fixed schedule is not known. Maybe the tolerability is the reason for varying the schedule. However that seems to lead to the exclusion of large proportion of the continuous group. This selects the fittest patients and causes a misbalance. The figures are poorly explained: free from progression and overall survival appears better for the "fixed schedule group" although this is not written in the text. The flow diagram of duration of response seems only for the 16 patients of the fixed group, the triangles and the stars are not explained. Finally the higher incidence of infections in the fixed group is not explained, since the CRS and ICANS were not different. A valid study should include the given dose and schedule and the reasons for drop outs.
Response 1:
We thank the reviewer for this comment and have clarified the rationale in the Introduction (page 2, line 16).
“However, while recent real-world and pharmacokinetic data indicate that de-escalation to biweekly dosing maintains disease control and reduces infection burden and treatment fatigue, no randomized or head-to-head studies have yet compared weekly, biweekly, and fixed-duration regimens.”
Comment 2:
The study limitations should be clearly stated.
Response 2:
Addressed in the newly added Discussion paragraph (page 11, line 5), highlighted in yellow (see Reviewer 1 Response).
Comment 3:
The figures are poorly explained: progression-free and overall survival appear better for the fixed group, though not stated in text.
Response 3:
We have revised the Results section for clarity and added explicit descriptions of the Kaplan–Meier curves (page 5, line 8).
“After a median follow-up of 12 months (IQR 6.0–15.0), the median progression-free survival (PFS) was 16 months in the continuous cohort and 13 months in the fixed cohort (HR 1.7; 95% CI 0.77–3.75; p = 0.20). The corresponding 12-month PFS estimates were 65% (95% CI 54–77%) and 66% (95% CI 45–96%) for the continuous and fixed groups, respectively. The median overall survival (OS) was not reached in the continuous cohort compared with 14 months in the fixed cohort (HR 2.79; 95% CI 0.96–8.07; p = 0.048). Although Kaplan–Meier curves visually appeared to favor the fixed-duration cohort for early disease control, these differences did not reach statistical significance.”
Figure legends for Figures 1–2 were also updated for clarity (pages 18–19).
Comment 4:
The flow diagram of duration of response lacks explanation for the symbols.
Response 4:
We have updated the Figure 3 legend (page 20, line 5) for clarity.
Comment 5:
Please provide the inclusion/exclusion criteria, dosing definitions, and reasons for treatment discontinuation.
Response 5:
We have revised the Methods section (page 3, line 5) to include this information.
“Patient selection followed criteria adapted from the pivotal MajesTEC-1 and MajesTEC-2 trials. Eligible patients were adults (≥18 years) with confirmed multiple myeloma per IMWG criteria and measurable disease (serum M-protein ≥1 g/dL, urine M-protein ≥200 mg/24 h, or abnormal free light chain ratio ≥10 mg/dL). All had relapsed or refractory MM after prior exposure to ≥1 proteasome inhibitor and ≥1 IMiD. Major exclusions included prior BCMA-targeted therapy, recent live vaccination, high-dose corticosteroids within 14 days, active CNS involvement, or HIV seropositivity.
Teclistamab was administered according to institutional practice consistent with the FDA-approved regimen and MajesTEC-1 protocol: subcutaneous step-up doses of 0.06, 0.3, and 1.5 mg/kg on days 1, 4, and 7, followed by 1.5 mg/kg once weekly (QW). In patients achieving a confirmed CR for ≥6 months, the dosing interval extended to every 2 weeks (Q2W).
The fixed-duration approach, applied in real-world practice, involved discontinuation after achieving ≥VGPR or in cases of toxicity or declining quality of life. After discontinuation, patients were monitored monthly until progression.”
Comment 6:
Explain why the fixed group had higher infection incidence despite similar CRS/ICANS rates.
Response 6:
We have added a mechanistic explanation in the Discussion (page 7, line 10):
“Although CRS and ICANS rates were comparable, the higher infection incidence in the fixed-duration group likely reflects prolonged immune suppression rather than acute cytokine toxicity. Teclistamab and other BCMA-targeted bispecific antibodies induce sustained B-cell depletion and hypogammaglobulinemia, predisposing to infections. Prolonged cytopenias, particularly neutropenia and lymphopenia, can persist beyond treatment discontinuation, increasing vulnerability. The fixed-duration cohort also had more prior therapy lines and longer follow-up, extending infection exposure. Differences in prophylactic interventions such as IVIG or antimicrobials may have contributed further.”

Reviewer 3 Report
Comments and Suggestions for Authors
The communication entitled: “Teclistamab Dosing Strategies in Relapsed/Refractory Myeloma: A Real-World Comparison of Weekly and Biweekly versus Fixed Intervals
ID: cancers-3912552ID.” by Snyder et al. investigated fixed dose teclistamab compared to continuous weekly or biweekly dosing in RRMM patients.
The manuscript is well written and of interest, therefore comments should be addressed to further improve the manuscript.
- Please highlight more intensively the clinical decision making process according different strategies.
- Result and discussion section: the authors mentioned that fixed duration of treatment is feasible and potentially effective on a selected patient population. Please highlight more deeper how clinicians should select patients for this (e.g. which patients would profit for this approach).
- Figures 1a, b, should be adapted since within the legends (see continuous vs. fixed) numbers are missing.
Author Response
Reviewer 3
Comment:
The manuscript is well written and of interest, therefore comments should be addressed to further improve the manuscript.
- Please highlight more intensively the clinical decision making process according different strategies.
- Result and discussion section: the authors mentioned that fixed duration of treatment is feasible and potentially effective on a selected patient population. Please highlight more deeper how clinicians should select patients for this (e.g. which patients would profit for this approach).
- Figures 1a, b, should be adapted since within the legends (see continuous vs. fixed) numbers are missing.
Response 1:
We thank the reviewer for this comment. We expanded the Discussion (page 8, line 6) to clarify the rationale for each teclistamab dosing approach:
“In clinical practice, the decision between continuous, biweekly, and fixed-duration teclistamab administration is individualized according to disease response, treatment tolerance, and patient preference. Continuous weekly dosing remains the standard of care and is preferred for patients with ongoing disease activity or partial response. Transition to a biweekly schedule is generally reserved for patients achieving a confirmed complete response sustained for at least six months, in line with FDA labeling.”
Response 2:
We further expanded the Discussion (page 8, line 13) to include patient-selection considerations:
“Selection of patients for a fixed-duration teclistamab strategy must be cautious and individualized. The most suitable candidates may be those who achieve sustained deep responses (≥VGPR) without biochemical relapse and who demonstrate recovery of immune parameters such as stable or increasing immunoglobulin levels or lymphocyte counts. This approach may also be considered for patients experiencing persistent cytopenias, recurrent infections, or treatment fatigue despite ongoing remission. Patients with aggressive disease or high-risk cytogenetics should generally continue continuous dosing until progression. These considerations are supported by emerging real-world experience with teclistamab, underscoring the complexity of immune recovery and infection risk after prolonged therapy.”
Response 3:
We have updated the figure legends accordingly (pages 18–20):
